# Chemogenetic generation of hydrogen peroxide in the heart induces severe cardiac dysfunction

Benjamin Steinhorn [1], Andrea Sorrentino [1], Sachin Badole [1], Yulia Bogdanova[2], Vsevolod Belousov[2,3,4] & Thomas Michel[1]

Oxidative stress plays an important role in the pathogenesis of many disease states. In the heart, reactive oxygen species are linked with cardiac ischemia/reperfusion injury, hypertrophy, and heart failure. While this correlation between ROS and cardiac pathology has been observed in multiple models of heart failure, the independent role of hydrogen peroxide ($H_2O_2$) in vitro and in vivo is unclear, owing to a lack of tools for precise manipulation of intracellular redox state. Here we apply a chemogenetic system based on a yeast D-amino acid oxidase to show that chronic generation of $H_2O_2$ in the heart induces a dilated cardiomyopathy with significant systolic dysfunction. We anticipate that chemogenetic approaches will enable future studies of in vivo $H_2O_2$ signaling not only in the heart, but also in the many other organ systems where the relationship between redox events and physiology remains unclear.

[1] Department of Medicine, Division of Cardiology, Brigham and Women's Hospital, Harvard Medical School, 75 Francis Street, Boston, MA 02115, USA.
[2] Shemyakin-Ovchinnikov Institute of Bioorganic Chemistry, Russian Academy of Sciences, GSP-7, Ulitsa Miklukho-Maklaya, 16/10, Moscow, Russia 117997.
[3] Pirogov Russian National Research Medical University, Moscow, Russia 117997. [4] Institute for Cardiovascular Physiology, Georg August University Göttingen, D-37075 Göttingen, Germany. These authors contributed equally: Benjamin Steinhorn, Andrea Sorrentino. Correspondence and requests for materials should be addressed to T.M. (email: thomas_michel@hms.harvard.edu)

The roles of reactive oxygen species (ROS) in biology continue to be wrapped in controversy despite decades of concerted investigation[1,2]. Long viewed as drivers of pathology and tissue degeneration, increased ROS and oxidative stress have been linked to diseases ranging from Alzheimer's disease to asthma to heart failure[3–5]. Indeed, it is difficult to open a pathology textbook and find a disease in which oxidative stress has not been claimed to play a role in its pathogenesis. These associations were originally made by observing that markers of oxidation are elevated in many disease states[6,7], and in vitro experiments have demonstrated a strong correlation between changes in redox stress and pathologic cellular responses[8,9]. However, the pathobiology of these ROS-associated diseases is complex, and the correlation of increased oxidation with disease does not necessarily imply direct causation. The causative relationship between increases in intracellular ROS and the pathologies with which they are associated has remained unclear, owing to the lack of available tools to precisely and specifically manipulate ROS production both in vitro and in vivo. The true independent effect of acute and chronic changes in intracellular ROS production is almost entirely unknown.

We chose to investigate the effect of the stable ROS hydrogen peroxide ($H_2O_2$) on the heart. Because of the heart's continual energetic demands supplied by oxidative phosphorylation, the heart is one of the most oxidatively active organs in the body, and the majority of cardiac pathologies have been associated with deranged oxidant signaling including hypertrophy, ischemia, and heart failure[10–12]. Given the significant global burden of cardiovascular disease, there has been intense interest in elucidating the roles of endogenous ROS production in the pathobiology of heart failure and heart disease. Most of the standard experimental models of heart failure have suggested that elevations in endogenous ROS production are correlated with cardiac dysfunction[13–17]. Despite the associations that have been observed between cardiac pathology and oxidative stress seen in these complex experimental systems, the field lacks a model that isolates oxidative stress as a unitary causative factor, and thus the effect of both acute and chronic increases in $H_2O_2$ in the heart independent of concomitant pathology is unknown. The work presented here addresses the fundamental unanswered question: what effect does intracellular $H_2O_2$ have on the heart?

To address this question, we apply a chemogenetic system for controlling intracellular $H_2O_2$ production based on a D-amino acid oxidase (DAAO) from the yeast *R. gracilis*, which catalyzes the conversion of D-amino acids to their corresponding alpha-keto acids, producing $H_2O_2$ in the process[18]. We find that delivery of a fusion protein between the $H_2O_2$-sensitive fluorescent biosensor HyPer[19] and DAAO via an adeno-associated virus type 9 vector achieves robust expression in the hearts of rats. In vitro stimulation of cardiac myocytes from these animals with D-alanine induces a significant transcriptional stress response with increased expression of targets of the transcription factors Nrf2 and NF-κB. In vivo activation of DAAO through the addition of D-alanine to the animals' drinking water results in severe systolic dysfunction over several weeks, associated with dephosphorylation of phospholamban and decreased expression of alpha myosin heavy chain. Our findings demonstrate that in vivo $H_2O_2$ generation in cardiac myocytes through chemogenetics induces a state of systolic heart failure without fibrotic remodeling. This application of chemogenetics in vivo lays the foundation for a chemogenetic approach that enables future studies of $H_2O_2$ signaling not only in the heart, but also in the many other organ systems where the transition from physiologic to pathologic redox signaling remains unclear.

## Results

**Development of the cardiac-specific HyPer-DAAO fusion protein**. Much in the way that optogenetic methods allow an experimenter to precisely elicit action potentials in neurons with an inert optical stimulus[20], chemogenetics enable one to reversibly activate an enzyme through the addition and removal of the enzyme's specific substrate. Chemogenetic approaches have previously been applied in vitro both to elicit physiologic and pathologic responses in cultured neurons and cancer cell lines[21,22]. For the experiments shown here, DAAO was fused to the C terminus of HyPer, a ratiometric $H_2O_2$-sensitive fluorescent biosensor[19]. The resultant fusion protein allows for simultaneous production (DAAO) and measurement (HyPer) of $H_2O_2$. The HyPer-DAAO fusion construct was then targeted to the cytoplasm by adding a nuclear exclusion sequence to the C terminus. In order to maximize cardiac specificity of expression in vivo, the construct was driven by a fusion between the CMV enhancer and the cardiac troponin T (cTnT) promoter[23] and delivered to rats intravenously in an adeno-associated virus serotype 9 vector. This system achieved robust and highly specific expression of the HyPer-DAAO fusion protein (DAAO) in the heart compared to other visceral organs—although we did detect modest expression of the construct in the skeletal muscle (Fig. 1c).

After injecting rats with AAV9 carrying the DAAO construct, cardiac myocytes were isolated 4–5 weeks later and imaged with live-cell fluorescence microscopy, taking advantage of the HyPer biosensor portion of the fusion protein to detect $H_2O_2$ production. Upon addition of D- but not L-alanine, $H_2O_2$ levels rose over the course of 45 min and plateaued, as measured by changes in HyPer fluorescence (Fig. 1a, b, Supplementary Figure 2); this rate of increase in response to D-alanine was dose dependent (Fig. 2a). By mutating one of HyPer's $H_2O_2$-sensitive cysteine residues to serine (resulting in the $H_2O_2$-insensitive probe named "SypHer"; reference 19), we have previously shown in cultured cell lines that this fluorescence change represents a true increase in $H_2O_2$ concentrations rather than a pH change[24]. In order to verify that the changes in HyPer fluorescence we observed in cardiac myocytes also are not an artifact of changes in pH, cardiac myocytes derived from human-induced pluripotent stem (IPS) cells were transfected with cDNA coding for the fusion between SypHer2 and DAAO (SypHer2-DAAO) and stimulated with D-alanine. In contrast to IPS-derived cardiac myocytes transfected with HyPer-DAAO, no significant changes in fluorescence were observed in myocytes expressing SypHer2-DAAO treated with D-alanine (Fig. 2b). Because D-alanine has only ever been measured at sub-nanomolar concentrations in mammals[25], this system serves as a tractable experimental model for dynamically regulating intracellular $H_2O_2$ production in cardiac myocytes through the addition or removal of D-alanine.

**In vitro transcriptional responses to DAAO activation**. We next examined the effect of acute activation of DAAO on targets of the redox-sensitive transcription factors Nrf2 and NF-κB[26]. Nrf2 translocates to the nucleus upon oxidation and subsequent ubiquitination of KEAP1 in the cytosol and promotes the transcription of what are typically thought of as cardio-protective genes[27]. In contrast, NF-κB dissociates from IKK-β upon oxidation in the cytosol and promotes transcription of a inflammatory set of genes, which are believed to be deleterious to cardiac function[28]. Two hours after the addition of D-alanine, but not L-alanine to cultured myocytes isolated from DAAO-infected rats, the Nrf2 transcripts *Hmox1*, *Nqo1*, and *Sxn1* (coding for heme oxygenase-1, NADPH quinone oxidase-1 and sulfiredoxin-1, respectively), as well as the NF-κB transcripts *Il1b*, *Tnfa*, *Icam1*, and *Nos2* (coding for interleukin 1β, tumor necrosis factor-α,

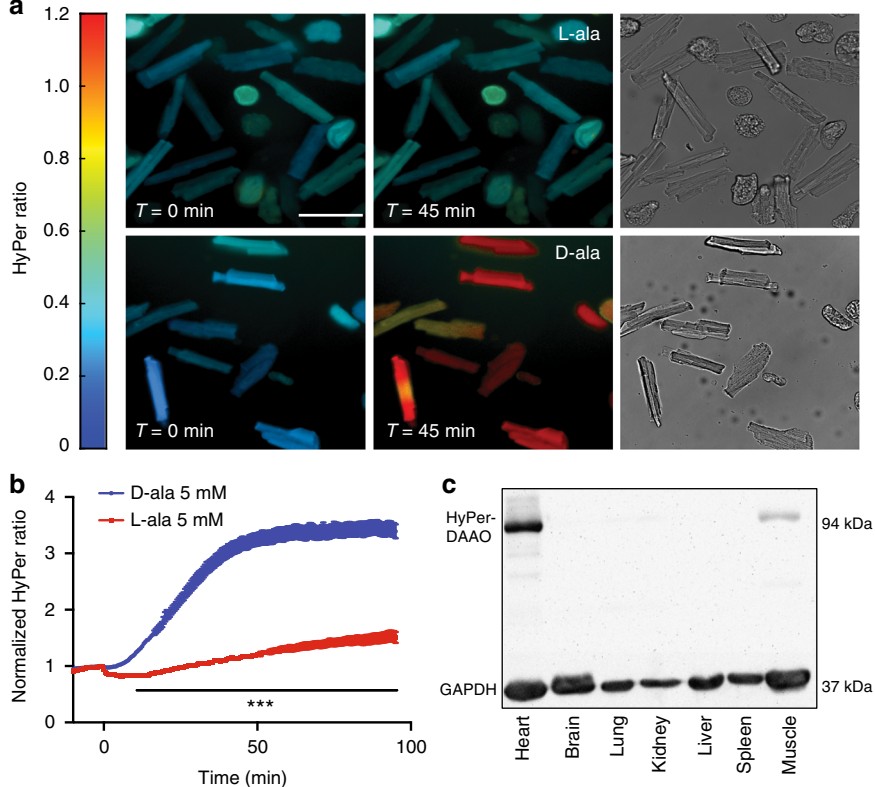

**Fig. 1** Expression and activation of the HyPer-DAAO fusion protein in cardiac myocytes. **a** Representative real-time ratiometric fluorescent images of cardiac myocytes isolated from rats infected with an AAV9 construct expressing the HyPer-DAAO fusion protein (DAAO). 4 weeks after infection, isolated myocytes were treated with 10 mM D- or L-alanine for the indicated times, and the HyPer ratio was quantitated. Images are displayed with a lookup table in which the color maps to the ratios shown in the colorbar, and the luminance maps to the intensity of HyPer's 420 nm excitation. Additional representative images of myocytes treated with D- vs. L-alanine can be found in Supplementary Figure 2 Scale bar, 50 μm. **b** Pooled ratiometric data from multiple myocytes treated with 5 mM D-alanine (blue circles, $n = 32$ cells) or L-alanine (red squares, $n = 21$ cells) for 90 min. Ratios are normalized to the HyPer ratio prior at time $t = 0$. ***$p < 0.001$ by two-way ANOVA with Bonferroni correction for multiple comparisons for D- vs. L-alanine. **c** Immunoblot probed for DAAO expression in lysates prepared from tissues isolated from rats infected with AAV9-DAAO. Lysates were probed with an antibody directed against YFP, which recognizes the HyPer component of the HyPer-DAAO fusion construct. This blot is representative of three experiments that yielded similar results. Data are represented as mean ± standard error

intracellular adhesion molecule 1, and inducible nitric oxide synthase) were significantly increased (Fig. 2c). Transcripts for the redox-active enzymes *Prdx1, Txn1*, and *Gpx3* (coding for the proteins peroxiredoxin-1, thioredoxin-1, and glutathione peroxidase-3) were also significantly elevated (Fig. 2d). This strong activation of both inflammatory as well as adaptive stress response elements indicates that the acute generation of intracellular $H_2O_2$ with DAAO induces a state of oxidative stress in cardiac myocytes. We note that while these data demonstrate evidence of robust activation of Nrf2 and NF-κB, short term transcriptional responses do not always accurately reflect the degree of change in protein levels due to potential concomitant changes in post transcriptional and translational regulatory mechanisms.

**Activation of DAAO in vivo**. After establishing that this chemogenetic system allows for rapid increases in $H_2O_2$ production in vitro that are sufficient to induce a stress transcriptional response, we next sought to determine the effect of chronic $H_2O_2$ production in vivo. Rats were injected with an AAV9 carrying the HyPer variant HyPer3 (control virus) or with the HyPer-DAAO fusion construct driven by the cTnT promoter (DAAO virus, described above), and both groups were fed D-alanine in their drinking water for 4 weeks. DAAO-expressing animals developed

severe systolic dysfunction with a reduced ejection fraction compared to control animals after two weeks of D-alanine treatment, and a dilated cardiomyopathy evolved over the course of 4 weeks as evidenced by increased end-diastolic left ventricular volumes (Fig. 3a, b). As an additional control for the effect of DAAO expression independent of activation by D-alanine, we fed L-alanine to DAAO-expressing rats for 5 weeks and observed no decrease in cardiac function (Supplementary Fig. 4). We next directly catheterized the left ventricle to better characterize the hemodynamics of this model of chronic intracellular $H_2O_2$ production. We found that the left ventricular systolic pressures and dP/dt during isovolumetric contraction were significantly lower in DAAO-expressing animals with no significant differences in end-diastolic pressures (Fig. 3c, d).

To further examine this profound decrease in systolic function with chronic DAAO activation in vivo, cardiac papillary muscles were isolated from DAAO-expressing and control animals following 4–5 weeks of treatment with D-alanine and studied ex vivo. Papillary muscles from animals expressing DAAO developed significantly lower systolic tension with maximal diastolic loading and responded only modestly to beta-adrenergic agonist stimulation with isoproterenol (Fig. 4a–e). Consistent with this beta agonist-unresponsive state, hearts from DAAO-expressing animals demonstrated a significantly reduced basal phosphorylation of phospholamban at both the serine 16

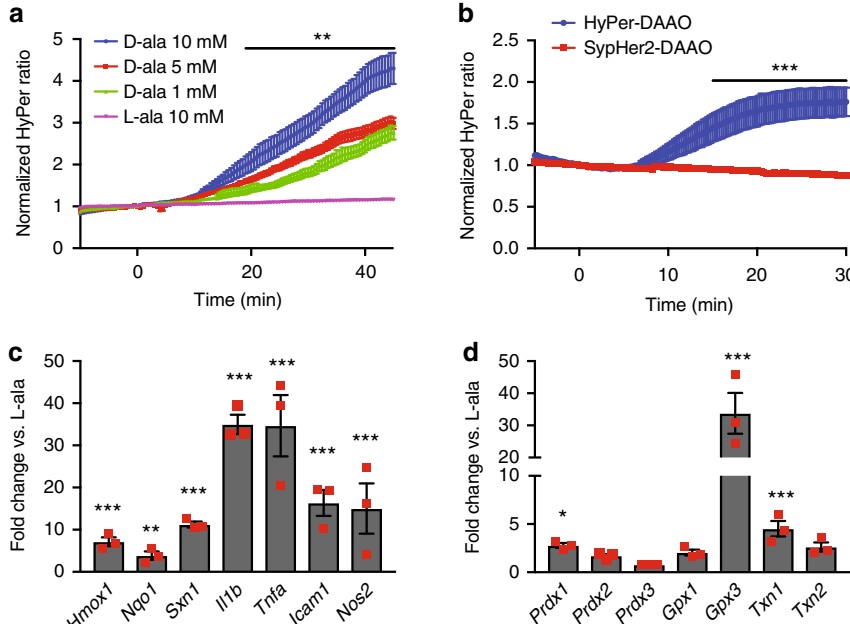

**Fig. 2** Fluorescence time course of DAAO activation and in vitro transcriptional responses. **a** Pooled ratiometric fluorescent data from myocytes treated with 1 mM (green triangles, $n = 20$ cells), 5 mM (red squares, $n = 20$ cells), or 10 mM (blue circles, $n = 9$ cells) D- alanine or 10 mM L-alanine (purple inverted triangles, $n = 14$ cells) for 45 min. Ratios are normalized to the HyPer ratio prior at time $t = 0$. **$p < 0.01$ by two-way ANOVA with Bonferroni correction for multiple comparisons for L- vs. D-alanine. **b** Ratiometric fluorescence time courses of human IPS-derived cardiac myocytes expressing HyPer-DAAO (blue circles, $n = 9$ cells) or SypHer2-DAAO (pH control construct; red squares, $n = 13$ cells) treated with D-alanine (10 mM). ***$p < 0.001$ by two-way ANOVA with Bonferroni correction for multiple comparisons. **c** Changes in expression of the Nrf2 transcriptional targets *Hmox1*, *Nqo1*, and *Sxn1* and the NF-κB transcriptional targets *Il1b*, *Tnfa*, *Icam1*, and *Nos2* in cardiac myocytes isolated from rats expressing DAAO and then treated with 10 mM D- vs. L-alanine for 120 min. Distributions for control samples treated with L-alanine can be found on Supplementary Figure 3A. **d** Relative changes in expression of the redox-active enzymes *Prdx1*, *Prdx2*, *Prdx3*, *Gpx1*, *Gpx3*, *Txn1*, and *Txn2* in response to 10 mM D- vs. L-alanine, analyzed in cardiac myocytes isolated from rats infected with DAAO. Distributions for control samples treated with L-alanine can be found on Supplementary Figure 3B. *$p < 0.05$, **$p < 0.01$, and ***$p < 0.001$ D- vs. L-alanine by ANOVA. Data are represented as mean ± standard error

and threonine 17 residues compared to control animals (Fig. 4f, g). While significantly different between the two groups, we did observe heterogeneity in phospholamban phosphorylation responses, which we attribute in part to variability in efficiency of retrograde perfusion and time taken to harvest samples. Ischemia and changes in extracellular calcium concentration are known to rapidly alter phospholamban phosphorylation[29]. We also found that expression of the fast twitch alpha myosin heavy chain protein (*Myh6*) was dramatically reduced in hearts from animals expressing DAAO, with no change in beta myosin heavy chain expression (*Myh7*, Fig. 4h). This significant reduction in alpha MHC expression—along with the beta-adrenergic unresponsiveness of hearts from DAAO-expressing animals—is consistent with the severe impairment of systolic function that we observed in vivo. We tested whether heterogeneity of physiologic and signaling responses in animals expressing DAAO could be explained by variability in efficiency of HyPer-DAAO expression by immunoblotting cardiac lysates for YFP, which is present in HyPer (Supplementary Figures 1A and 1B). While there was variability in DAAO expression, we did not observe any clear correlation with severity of the phenotype induced by chronic exposure to D-alanine.

**Signaling responses to in vivo DAAO activation**. To better characterize the cardiac dysfunction induced by activation of DAAO in vivo, we looked for evidence of heart failure by measuring expression of atrial and B-type natriuretic peptide (ANP and BNP, Fig. 4i, j). Expression of both ANP and BNP were significantly elevated in cardiac tissue from animals expressing DAAO. Serum BNP is a widely used clinical marker of heart

failure, and is associated with increased diastolic ventricular wall stress[30]. While both ANP and BNP protein levels were elevated in cardiac tissue (Fig. 5a), only plasma ANP was significantly increased in DAAO-expressing animals (Fig. 5b). However, plasma cardiac troponin I (cTnI), a widely used clinical metric of myocardial injury was significantly elevated in animals expressing DAAO vs. control animals chronically fed D-alanine (Fig. 5c). We next probed the hearts for transcriptional evidence of oxidative stress. Nrf2 activity was significantly increased in hearts from animals in which DAAO was chronically activated (Fig. 6a). However, in contrast to the transcriptional responses we observed with acute in vitro $H_2O_2$ production (Fig. 1), neither NF-κB activity nor expression of members of the peroxiredoxin, thioredoxin, or glutathione peroxidase families were significantly elevated in this model of chronic in vivo $H_2O_2$ production (Fig. 6a, b).

In order to better understand the relevance of this model of cardiac oxidative stress to natural cardiac pathologies, we measured the relative degree of oxidation of the glutathione pool in the hearts of rats expressing DAAO. The highly abundant tripeptide glutathione ("GSH") provides much of the cell's reductive power. It aids in the reduction of oxidized proteins and serves as a sink for $H_2O_2$ in a process catalyzed by glutathione peroxidase, which consumes two glutathione molecules to yield glutathione disulfide ("GSSG"). The concentration of reduced GSH was decreased in the hearts of animals expressing DAAO by ~2 fold (Fig. 6c), a reduction that is similar to what has been reported in animal models of heart failure including myocardial infarction and diabetic cardiomyopathy[7,31]. Similarly, the total amount of available glutathione (reflecting the sum of

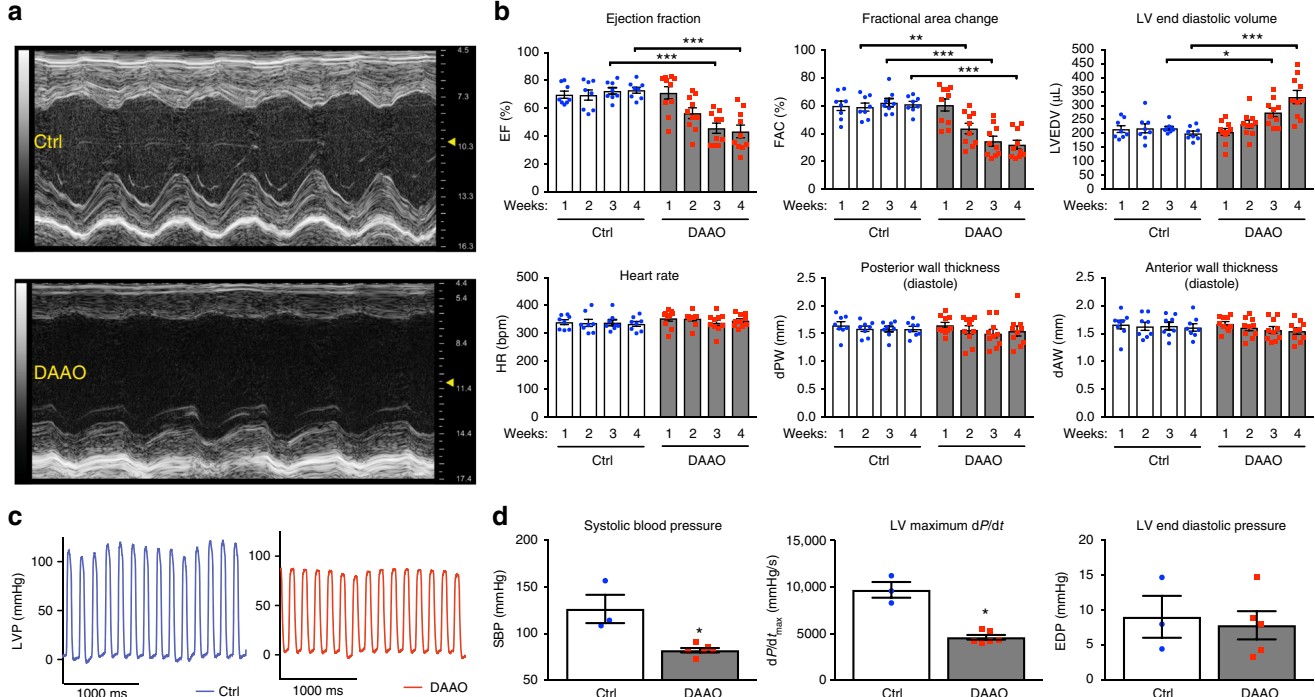

**Fig. 3** Echocardiographic and intracardiac pressure parameters after chronic activation of DAAO. **a** Representative M-mode images from short-axis views of the left ventricle of rats infected with control (Ctrl) or DAAO virus and treated with D-alanine in their drinking water for 4 weeks. **b** Left ventricular echocardiographic parameters over the 4 weeks of oral D-alanine treatment in rats expressing DAAO (red squares) vs. control animals (blue circles). *$p < 0.05$, **$p < 0.01$, and ***$p < 0.001$ by ANOVA. **c** Representative left ventricular (LV) pressure traces of Ctrl and DAAO-infected rats after 4 weeks of oral D-alanine treatment. **d** Pooled LV pressure parameters after 4 weeks of treatment. *$p < 0.05$ by Mann–Whitney $U$ test. Data are represented as mean ± standard error

GSH and GSSG levels) is increased in the hearts of animals expressing DAAO (Fig. 6d), which is consistent with the changes that have been observed in pressure overload transverse aortic constriction models of rodent heart failure[16]. Both the decrease in available GSH and increase in total glutathione suggest that the level of oxidative stress produced by DAAO in vivo are similar to the pathologic levels that have been observed in other animal models of human heart failure. Chemogenetic approaches using DAAO allow the independent effects of oxidative stress on cardiac physiology to be directly tested, without the potential confounding effects of more complexly-determined disease processes such as myocardial infarction, metabolic syndrome, and aging.

To further investigate the systolic dysfunction that occurs with long term intracellular $H_2O_2$ generation in the heart, we tested hearts from animals chronically treated with D-alanine for myocyte dropout and cardiac fibrosis, both of which have been attributed to oxidative stress in various other experimental models of heart failure, including transverse aortic constriction and myocardial infarction[17]. We analyzed histological sections and observed no architectural differences between hearts from DAAO-expressing and control animals or evidence of myocyte dropout (Fig. 4g). These findings are consistent with the lack of any significant changes in wall thickness observed by echocardiography (Fig. 3b) or in the abundance of the sarcomeric protein cTnT (Fig. 4f). Staining of cardiac tissue sections with Masson's trichrome stain also showed no differences in fibrotic remodeling (Fig. 6g, h). This finding was supported by our observation that levels of collagen, matrix metalloprotease 2, galectin-3, and TGF-β expression—all of which are markers of tissue fibrosis[30]—were similar in cardiac tissue from both groups (Fig. 6e, f). These biochemical and histologic data demonstrate that chronic oxidative stress caused by DAAO can induce profound

ventricular dysfunction without causing fibrotic remodeling in the heart, a histologic change that has been observed in most experimental models of heart failure associated with excess ROS production[17]. We therefore speculate that fibrosis may be a later consequence rather than an early causal factor in the development of heart failure.

## Discussion

The work presented here introduces an informative application of chemogenetics to the in vivo setting for studying redox signaling. However, we recognize several limitations to this system. It is important to note that D-amino acids are present in some mammalian systems, albeit at picomolar concentrations[25]. The exact physiologic role of these endogenous D-amino acids remains unclear, but they are nonetheless present at far lower concentrations than what is needed to activate the recombinant DAAO used for these experiments. This conclusion is supported by the observation that animals expressing DAAO had no change in cardiac function when fed L-alanine (Supplementary Figure 5). Similarly, mammalian isoforms of DAAO have been described in the brain and kidney, but their kinetics are several orders of magnitude slower than the recombinant yeast DAAO[18,32]. We also note that in DAAO catalysis, ammonia ($NH_3$) is produced in equimolar amounts as $H_2O_2$. However, ammonia is present in cells at much higher concentrations than $H_2O_2$, and the relative increase in intracellular $NH_3$ following DAAO activation is therefore nominal[33]. In addition, the cellular effects of $NH_3$ on cells occur at far higher concentrations than $H_2O_2$, with effects of $NH_3$ on metabolic pathways being observed starting at 0.5 mM[34]. This is in contrast to $H_2O_2$ that can elicit cellular responses in the heart at micromolar concentrations[35]. We therefore conclude that the cellular effects of DAAO reflect the generation of $H_2O_2$ and not $NH_3$.

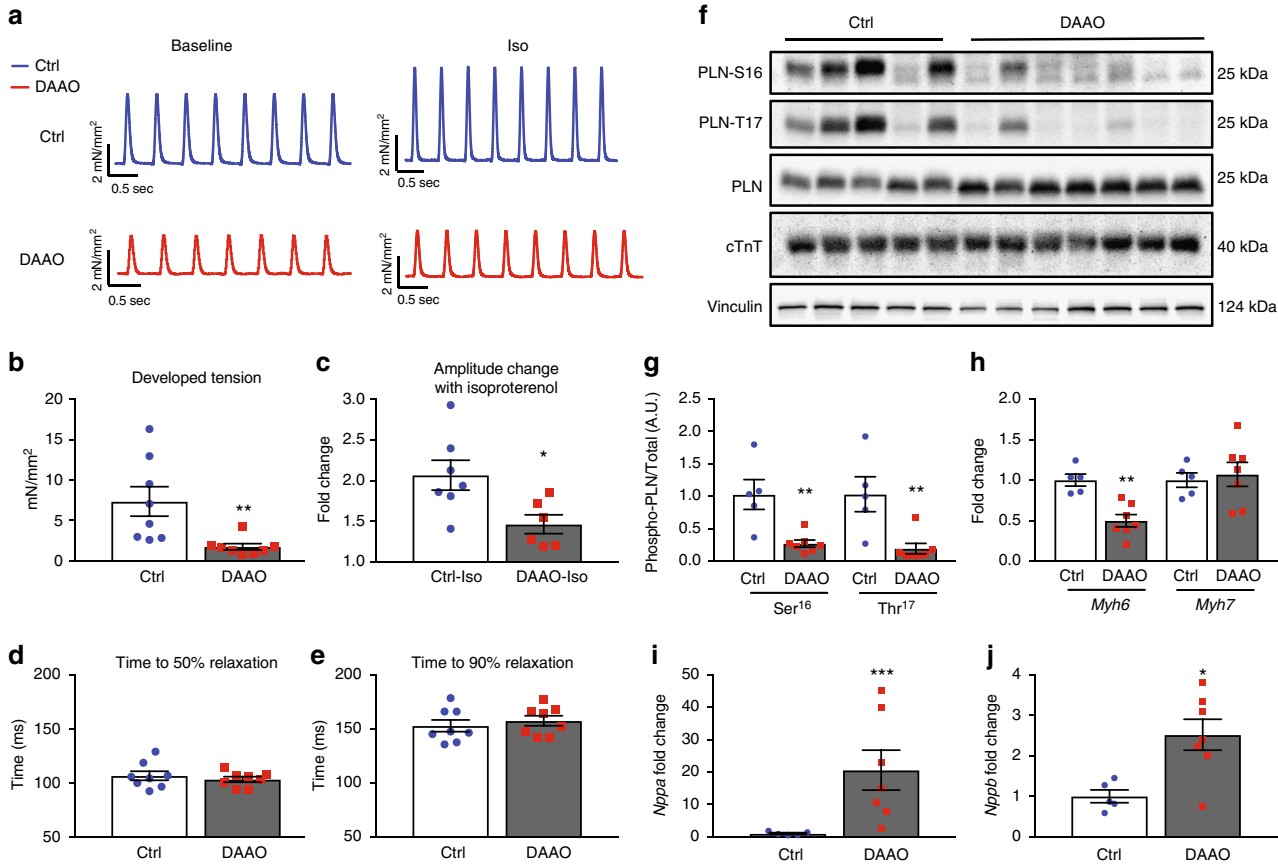

**Fig. 4** Effects of chronic in vivo DAAO activation on ex vivo cardiac responses and signaling. **a** Representative traces of tension generated by paced papillary muscles isolated from control animals or animals expressing DAAO and treated with oral D-alanine for 4 weeks, with and without isoproterenol stimulation (100 nM). **b**, **c** Isometric contractile force and **d**, **e** relaxation times for papillary muscles at baseline and after treatment with the beta-adrenergic agonist isoproterenol (100 nM). Isoproterenol contractile force is represented as the fold change from the baseline developed tension in the absence of agonist. *$p < 0.05$ and **$p < 0.01$ by t-test. **f** Immunoblots of cardiac lysates from Ctrl and DAAO-infected animals and probed for phospho-phospholamban (PLN) at phospho-serine 16 and phospho-threonine 17, and for total cardiac troponin T (cTnT). **g** Densitometry of immunoblots shown in **f**. **$p < 0.01$ by ANOVA. Immunoblotting of the same lysates for YFP, a component of HyPer, did reveal some heterogeneity in efficiency of expression from animal to animal, but did not correlate with signaling or physiologic responses (Supplementary Figures 1A and 1B). **h** Relative changes in expression of alpha MHC (*Myh6*) and beta MHC (*Myh7*) in hearts from DAAO (red squares) vs. Ctrl (blue circles) infected animals. **i** Relative changes in expression of atrial natriuretic peptide (*Nppa*) in hearts of control (Ctrl) and DAAO-expressing animals treated with oral D-alanine for 4 weeks. ***$p < 0.001$ by t-test. **j** Relative expression of B-type natriuretic peptide (*Nppb*) in hearts from Ctrl and DAAO animals. *$p < 0.05$ by t-test. Direct measurement of ANP and BNP protein levels by ELISA in cardiac lysates from control (Ctrl) and DAAO-expressing animals. *$p < 0.05$ by t-test. Data are represented as mean ± standard error

We anticipate that the work presented here will open the door to a better understanding of the roles of $H_2O_2$ both in normal cardiac physiology and in cardiac disease states. Our studies help to resolve the questions of causality that have undermined mechanistic interpretations of more complex animal models of heart failure: these chemogenetic approaches establish that oxidative stress alone is sufficient to cause cardiac dysfunction without fibrotic remodeling. We believe that chemogenetic approaches using DAAO can be more broadly exploited to test other long-held hypotheses in the field of cardiac redox biology. For example, it has been proposed that the divergent responses to intracellular $H_2O_2$ stem from differences in the subcellular localization of oxidants[36]. This hypothesis is now testable by targeting DAAO to various subcellular compartments including the mitochondria, plasma membrane, and nucleus. Likewise, dose-dependent effects of $H_2O_2$ can now be determined in vivo by altering the dose of D-alanine delivered to the animals. Chemogenetic approaches can also be applied to probe the in vivo role of $H_2O_2$ in many of the other organ systems where ROS have been implicated to play a part in physiology and/or disease. From

smooth muscle beds, where $H_2O_2$ is thought to regulate vascular tone[37] to the brain, where $H_2O_2$ has been suggested to play a role in neurodegeneration[4], it is our hope that chemogenetic approaches may help to clarify the mechanisms whereby ROS elicit pathological as well as physiological responses in diverse organ systems.

## Methods

**Construction and chemogenetic activation of DAAO.** The cytosolic (extra-nuclear) version of DAAO was created by PCR amplification of cDNA coding for DAAO (forward primer 5′-ACC GCT AGC GCC ACC ATG TCC GTC CTG ACG CC-3′; reverse primer: 5′-ACC AAG CTT TTA CAG GGT CAG CCG CTC CAG GGG GGG CAG GCT CTC CCT AGC TGC GC-3′), which added a nuclear exclusion sequence (NES) to DAAO. The PCR fragment was then ligated into the pC1-CMV vector. The HyPer-DAAO-NES and SypHer2-DAAO-NES constructs were generated by fusing the cDNA for HyPer or SypHer2 with DAAO-NES using the NEBuilder HiFi DNA assembly system (New England Biolabs). Primers were designed using the NEBuilder online assembly tool, and a Gly-Gly-Ser-Gly linker was introduced between HyPer and DAAO (HyPer/SypHer2 forward primer: 5′-GAT TCG AAC ATC GAT TGA ATT CGC CAC CAT GGA GAT GGC A-3′; Reverse primer 5′-TGT GAC CAC TAC CAC CAA CCG CCT GTT TTA AAA CTT TAT CG 3′. DAAO-NES forward primer: 5′- AGG CGG TTG GTG GTA GTG GTC ACA GCC AGA AGA GGG TG-3′; Reverse primer: 5′- GCG CTG CTC

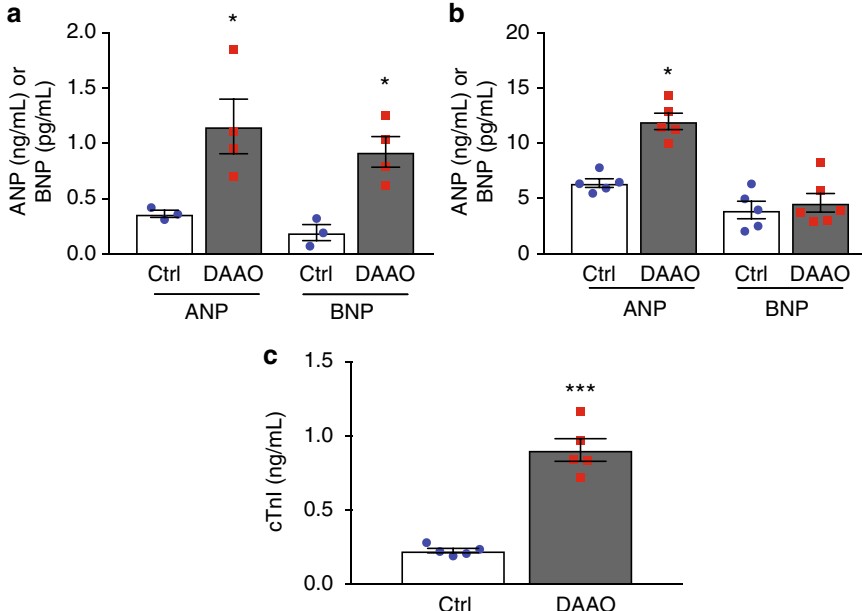

**Fig. 5** Changes in tissue and plasma cardiac biomarkers with chronic DAAO activation. **a** Tissue levels of atrial (ANP) and B-type (BNP) natriuretic peptide in cardiac lysates from control (Ctrl) and DAAO-expressing animals fed D-alanine for 4 weeks. *$p < 0.05$ by $t$-test. **b** Plasma levels of ANP and BNP in Ctrl and DAAO-expressing animals chronically fed D-alanine. *$p < 0.05$ by $t$-test. **c** Plasma levels of the cardiac biomarker troponin I (cTnI) in Ctrl and DAAO animals. ***$p < 0.001$ by $t$-test. Data are represented as mean ± standard error

GAG GCA AGC TTA CAG GGT CAG CCG CTC-3′). The resultant fusion product was inserted into the Stratagene CMV AAV expression vector between the EcoRI and HindIII restriction enzyme sites. The CMV promoter was replaced with the cTnT promoter [21] between the NcoI and SacI sites. HyPer3 was cloned into the Stratagene CMV AAV expression vector between the EcoRI and HindIII restriction sites by PCR to generate the control virus. Constructs were verified by whole plasmid sequencing (Massachusetts General Hospital DNA core, complete sequences in Supplementary Data 1) and packaged into AAV9 viral particles with the Rep2/Cap9 AAV encapsulation construct by the Children's Hospital Boston Viral Vector core. Approximately $10^{12}$ viral genome copies were suspended in 100 microliters phosphate-buffered saline and injected intravenously into 40–50 g (3–4-week-old) male Wistar rats (Charles River Labs). Oral D-alanine treatments and analyses of isolated cardiac myocytes were performed 4–6 weeks following viral infection, after which the rats were ~400 g in weight. For in vivo chronic activation of DAAO, the rats' drinking water was supplemented with D-alanine (1 M, Oakwood Chemical). Body weights and common clinical serum biomarkers (assayed by the Brigham and Women's Hospital clinical pathology laboratory) were measured after chronic feeding (Supplementary Figure 8).

**Live-cell fluorescent imaging**. Following isolation, adult ventricular cardiac myocytes were plated on glass dishes (Mattek) and maintained in Hanks' balanced salt solution in preparation for live-cell fluorescence imaging[38]. The dishes were then mounted on an Olympus DSU inverted fluorescence microscope and maintained at room temperature. IPS-derived cardiac myocytes were plated on 13 mm Thermanox plastic coverslips (Nunc) coated with fibronectin (Sigma) in order to optimize cell adhesion. IPS-derived cardiac myocytes were incubated with 10 μM flavin adenine dinucleotide (FAD, Sigma) for at least an hour prior to imaging. For live-cell imaging, HyPer and SypHer2 were excited at 420 nm and 480 nm while visualized with either a ×20 (Adult ventricular myocytes) or ×40 (IPS-derived myocytes) oil immersion objective (Olympus). Images were acquired with a CCD camera (Hammamatsu) after filtering at HyPer's emission wavelength of 530 nm. Images were analyzed post-hoc with a custom-written macro in Metamorph software (Olympus). IPS-derived cardiac myocytes images were acquired and analyzed in Metafluor software (Olympus). After background subtraction, ratiometric images were generated by dividing images excited at 480 nm by images excited at 420 nm. Cardiac myocytes which detached from the coverslip or lost membrane integrity (as evidenced by contraction of the myocyte into a rounded shape) were excluded from analysis. Fluorescent ratio traces were obtained by measuring the average ratio in a region of interest placed over each cardiac myocyte. In order to mitigate artifact from any spontaneous contractions during the time-course experiments, ratiometric time-course traces were median filtered with a window size of three frames prior to statistical analysis. Ratiometric images with colorbar were generated with the scripting language MATLAB (Mathworks). Ratiometric images are displayed with a color lookup table in which the color corresponds to the HyPer ratio, and the luminance corresponds to the fluorescent intensity of the 420 nm excitation (the denominator in the calculated ratio).

**Isolation and culture of adult rat cardiac myocytes**. All animal experimentation was performed according to protocols approved by the Brigham and Women's Hospital Committee on Use of Animals in Research, and these studies have complied with all relevant ethical regulations. 10–14-week-old male Wistar rats were anesthetized with isofluorane, heparinized (500 U ip), sacrificed, and cardiac myocytes were then isolated[39]. The heart was quickly removed from the chest and retrogradely perfused through the aorta with a calcium-free HEPES-buffered solution. The heart was then enzymatically digested with type 2 collagenase (Worthington, 7500 U in 50 mL) and protease XIV (Sigma, 4 mg in 50 mL) and the calcium concentration was steadily increased to 200 μM. The heart was minced and digested further with gentle agitation at 37 °C and trituration over 10 min. The concentration of calcium was gradually increased to 800 μM during agitation. Finally, myocytes were filtered and washed twice with a buffer containing 0.1% BSA and 1.25 mM calcium chloride prior to plating. Myocytes were plated on laminin-coated dishes at a density of 50,000–100,000 myocytes per 35 mm dish in M199 medium (Gibco) supplemented with 5% fetal bovine serum, penicillin/streptomycin, L-carnitine, creatine, and taurine. After 2 h, the media was switched to M199 supplemented with penicillin/streptomycin, L-carnitine, creatine, and taurine.

**Cell culture**. After isolation, rat ventricular myocytes were plated on laminin-coated 6-well culture dishes in plating medium containing M199 supplemented with L-carnitine, creatine, taurine, and 5% fetal bovine serum[39]. Following 1 h of incubation, the medium was switched to M199 supplemented with L-carnitine, creatine, taurine, and penicillin/streptomycin. Human IPS-derived cardiac myocytes were purchased from Ncardia (Plymouth Meeting, PA) and cultured according to their instructions. Briefly, the cells were thawed and directly plated onto 13 mm plastic Thermanox (Nunc) coverslips coated with fibronectin (Axol) and cultured in maintenance media from Ncardia. The day after plating, the cells were transfected with cDNA for either HyPer-DAAO-NES or SypHer2-DAAO-NES using Polyjet plasmid transfection reagent (Signagen). Cells were imaged the following day as described above.

**Echocardiography**. Echocardiographic images were acquired with a Visual Sonics Vevo 3100 system equipped with a MX250 (13–24 MHz) probe. Rats were lightly anesthetized with 1.5–2% isofluorane prior to recording of standard mid-papillary level short-axis and long-axis views of the heart. M-mode and anatomic images were analyzed in vivo-lab software. Both the sonographer and analyzer were blinded to the experimental group of the animals.

**LV pressure analysis**. Rats were anesthetized with ~3% isofluorane delivered by a nose cone. The right carotid was dissected from the surrounding structures with care taken to avoid damaging the vagus nerve. The rostral portion of the carotid was ligated with silk suture, and a SPR-869 (Millar) catheter was introduced into the carotid while proximal flow was occluded with a bulldog clamp. The clamp was

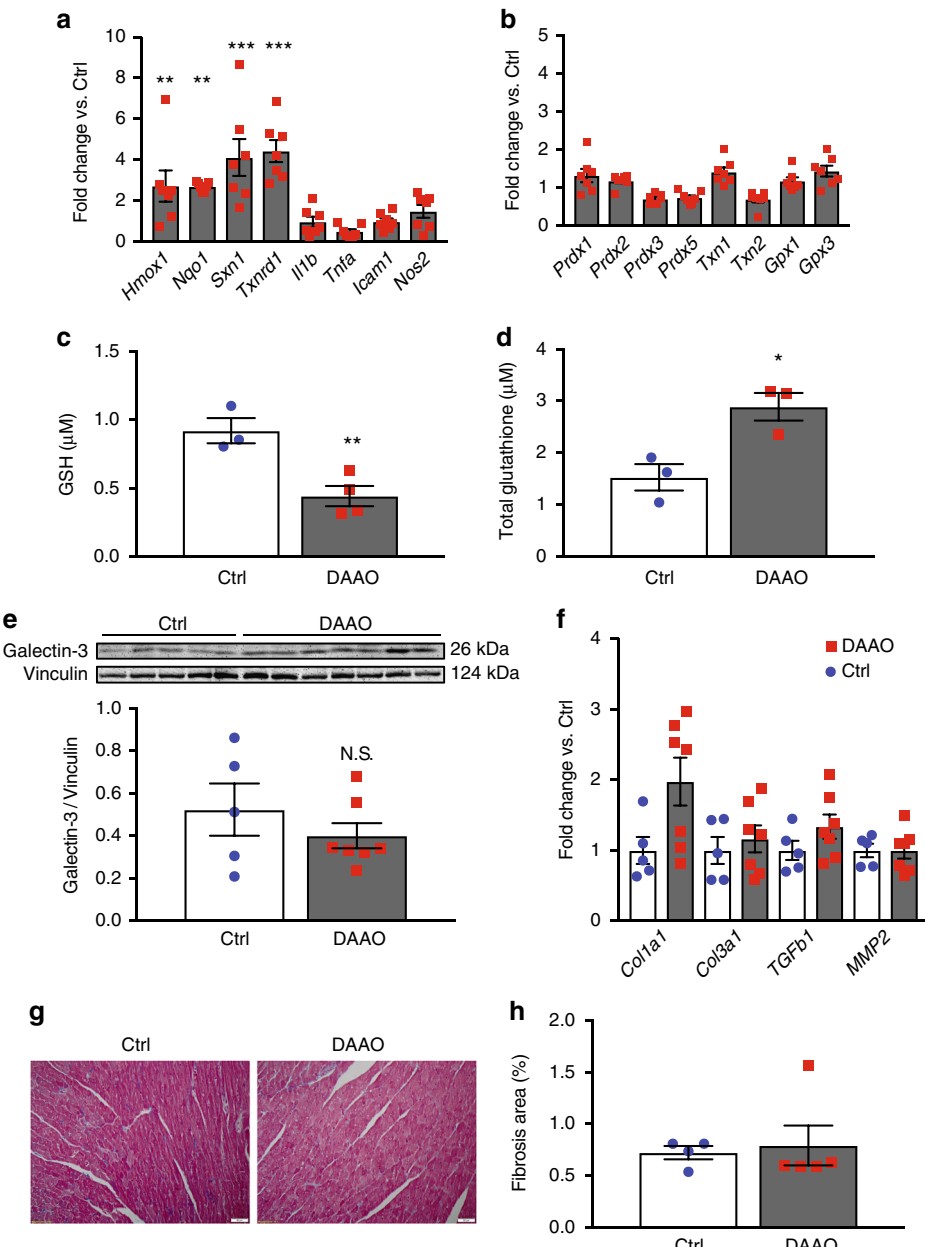

**Fig. 6** Changes in markers of oxidative stress and fibrosis after chronic activation of DAAO. **a** Relative changes in expression of the Nrf2 targets *Hmox1*, *Nqo1*, *Sxn1*, and *Txnrd1* and the NF-κB targets *Il1b*, *Tnfa*, *Icam1*, and *Nos2* in hearts from animals infected with control AAV9 (Ctrl) and DAAO virus measured by qPCR. Distributions for animals infected with control virus can be found on Supplementary Figure 4C. **p < 0.01 and ***p < 0.001 by ANOVA. **b** Relative changes in expression of the reductive enzymes *Prdx1*, *Prdx2*, *Prdx3*, *Gpx1*, *Gpx3*, *Txn1*, and *Txn2* in Ctrl and DAAO hearts measured by qPCR. Distributions for animals infected with control virus can be found on Supplementary Figure 4D. **c** Reduced (GSH) measured in hearts from animals expressing DAAO vs. Ctrl. **p < 0.01 by t-test. **d** Total glutathione measured in hearts from DAAO-expressing vs. control animals treated with D-alanine for 4 weeks. *p < 0.05 by t-test. **e** Immunoblot and densitometry for the fibrotic marker galectin-3 in cardiac lysates from Ctrl and DAAO animals. **f** Relative changes in expression of the fibrosis-associated transcripts *Col1a1*, *Col3a1*, *Tgfb1*, and *Mmp2* in hearts from Ctrl (blue circles) and DAAO (red squares) animals. No significant differences were observed by ANOVA. **g** Representative histology of hearts from Ctrl and DAAO animals stained for collagen with Masson's trichrome stain. **h** Masson trichrome-stained cardiac tissue sections quantitatively analyzed for fractional area of fibrosis. The images shown are representative of n = 4 animals from each group, which were pooled for statistical analysis. Data are represented as mean ± standard error

then removed, the catheter rapidly advanced and a suture was placed to prevent blood loss. The catheter was then advanced into the left ventricle. Pressure signals were recorded through an MPVS-400 analog to digital converter interface (Millar) and analyzed using Labchart version 8 software (ADInstruments).

**Ex vivo papillary muscle analysis.** Papillary muscles were dissected from the Rat LV and mounted in a horizontal tissue bath (Steiert, Hugo Sachs Elektronik-Harvard Apparatus) equipped with a force transducer (Harvard Apparatus)[40]. Muscles were superfused with Krebs–Henseleit buffer (KHB) solution containing

1.8 mM $CaCl_2$ at 37 °C, saturated with 95% oxygen and 5% $CO_2$. The myocardium was stimulated by two platinum electrodes via field stimulation (isolated stimulator output: frequency 2 Hz; pulse duration 2 ms; intensity 1.5-fold threshold; UISO, Hugo Sachs Elektronik-Harvard Apparatus). Each muscle was stretched to the length at which force of contraction was maximal. Muscle preparations were allowed to equilibrate for at least 30 min. Developed tension was measured isometrically with the force transducer attached to a bridge amplifier (ADInstruments) and a 4 kHz A/D converter (PowerLab 4/30, ADInstruments). Tension was recorded and analyzed with LabChart 8 Pro software (ADInstruments). Time to 50

and 90% relaxation are expressed with respect to the beginning of the twitch. Papillary muscles were fixed in 10% formalin and the cross-sectional area was measured (ImageJ software). Force measurements were then normalized to cross-sectional area (mN mm$^{-2}$).

**Immunoblot analyses**. Hearts were rapidly removed and perfused with ice-cold PBS. The left ventricle was dissected from the rest of the tissues, minced, and suspended in lysis buffer (50 mM Tris-HCl, pH 7.4; 150 mM NaCl; 1% Nonidet P-40; 0.25% sodium deoxycholate; 1 mM EDTA; 2 mM $Na_3VO_4$; 1 mM NaF; 2 μg/mL leupeptin; 2 μg/ml antipain; 2 μg/ml soybean trypsin inhibitor; and 2 μg/mL lima trypsin inhibitor) and snap frozen. The lysates were homogenized and centrifuged. With the exception of samples to be probed for phospholamban, cell lysates were then boiled with SDS (2%) and ß-mercaptoethanol (5%). After separation by SDS-PAGE, proteins were electroblotted onto nitrocellulose membranes. After incubating the membranes in 3% BSA in Tris-buffered saline with 0.1% (vol/vol) Tween 20 (TBST), membranes were incubated overnight in TBST containing 3% bovine serum albumin plus the specified primary antibody. The Galectin-3 antibody was diluted 1:500; all other primary antibodies were diluted 1:1000. Phospholamban antibodies were from Badrilla (phospho-serine 16: A010-12, phospho-threonine 17: A010-13 and total phospholamban: A010-14), and Galectin-3 was from Sino Biological (10289-R078). Vinculin (catalog #18799), cardiac Troponin T (catalog #5593), GAPDH (catalog #3683), and GFP (catalog #2956, recognizes GFP, YFP, and RFP) were from Cell Signaling Technology. After four washes (10 min each) with TBST, the membranes were incubated for one hour with a horseradish peroxidase-labeled goat anti-rabbit or anti-mouse immunoglobulin (Cell Signaling Technology) secondary antibody (1:5000 dilution) in TBST containing 3% BSA. The membranes were washed four additional times in TBST, then incubated with a chemiluminescent reagent according to the manufacturer's protocols (SuperSignal West Femto), and digitally imaged in a chemiluminescence imaging system (Alpha Innotech Corporation, San Leandro, CA). All unedited images of western blot membranes with molecular weights indicated can be found in Supplementary Figures 6–9.

**Histological analysis**. LV myocardial tissue was fixed in phosphate-buffered formalin (10%, Sigma) and embedded in paraffin. Masson's trichrome stain for detection of connective tissue was performed by the Pathology Core at the Brigham and Women's Hospital. Images were acquired with an upright microscope (Olympus BX63). Interstitial fibrosis was quantified as a percentage of total tissue area using ImageJ software, and the individual responsible for collecting images and analyzing the data was blinded to the experimental group of the samples.

**mRNA transcript quantification**. For in vitro experiments, cells were lysed in Trizol reagent (Invitrogen). For cardiac tissue, hearts were perfused with ice-cold PBS in a retrograde fashion through the aorta to remove any blood. The left ventricle was then dissected from the rest of the heart, minced, and suspended in Trizol and homogenized. RNA was isolated according to the manufactures' instructions. cDNA was generated with either the Protoscript II first strand cDNA synthesis kit (New England Biolabs) or the AzuraQuant cDNA synthesis kit (Azura Genomics). cDNA was amplified in a StepOneplus RT-PCR thermocycler (Applied Biosciences) with AzuraQuant Green Fast qPCR Master Mix (Azura Genomics). Genes of interest were amplified with gene-specific primers (see Supplementary Table 1 for primer sequences). *Hprt* was used as a house-keeping gene to which genes of interest were normalized.

**Quantification of ANP, BNP, and cTnI by immunoassay**. ANP and BNP levels in plasma and cardiac tissue were measured with rat BNP-45 and ANP ELISA kits (Abcam) according to the manufacturer's instructions. Cardiac troponin I (cTnI) was measured in plasma with a cTnI ELISA kit (Life Diagnostics).

**Quantification of reduced and total glutathione**. Reduced glutathione was measured from fresh cardiac tissue samples. After homogenization in ice-cold Tris buffer, protein concentrations were measured, and the samples were then deproteinized by trichloroacetic acid precipitation. The supernatant was then combined with 0.6 M 2-nitro benzoic acid in an 8:1 ratio to a final volume of 200 microliters, and absorption was measured at 412 nm. A standard curve was prepared by serial dilutions of reduced glutathione (Sigma). Total glutathione was assayed with a commercial glutathione fluorometric assay (Abcam) according to the manufacturer's instructions.

**Statistical analysis**. All experiments were performed at least three times. Mean values for individual experiments are expressed as means ± standard error. Distributions were analyzed for normality with the Shapiro–Wilk test. Comparisons between two groups were assessed by *t*-test if normally distributed or by Mann–Whitney *U* test if not normally distributed. For comparison of papillary muscle contractility pre- and post-isoproterenol treatment, a paired *t*-test was used. Comparisons between multiple groups, including quantitative qPCR and echo-cardiographic measurements, were analyzed by ANOVA with Sidak's multiple

comparison test. Fluorescence time-course experiments were analyzed by two-way ANOVA with a Bonferroni post-test. A *p* value of $< 0.05$ was considered statistically significant.Data availability. All datasets generated and analyzed in the current work are available from the corresponding author on reasonable request.

## Data availability

All datasets generated and analyzed in the current work are available from the corresponding author on reasonable request.

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

## Acknowledgements

The authors would like to thank Luca Troncone and Jana Bagarova for helpful conversations regarding molecular techniques and Tanni Arif for technical and administrative support. These studies were supported by NIH grants PO1-HL48743 and RO1-HL46457 (to T.M.), T32-GM007753 (B.S.), and T32-HL007604 (S.B.); Brigham and Women's Hospital Health and Technology Innovation Award (to T.M.); American Diabetes Association grant 9-17-CMF-012 (to A.S.); and Russian Science Foundation grant 17-14-01086 and DFG IRTG 1816 (to V.B.)

## Author contributions

B.S., A.S., S.B., V.B., and T.M. designed experiments. B.S., A.S., Y.B., and S.B. performed and analyzed experiments. B.S., A.S., V.B., and T.M. wrote the manuscript.

## Additional information

**Competing interests:** The authors declare no competing interests.

