## [Peer Review File · Nature Communications]

Reviewers' Comments:

Reviewer #1:

Remarks to the Author:

This is a stimulating study presenting some very interesting findings, making use of a technology probably only used before in cell systems. The data collected using the model in which peroxide can be measured and inducibly increased by provision of a D-amino acid is an interesting and potentially valuable approach that would be of broad interest to many research fields. As the authors conclude, this work provides evidence that such "chemogenetic" may help in our understanding of redox alteration to disease, in this case heart failure.

I am confident of the finding and a comprehensive series of experiments have been performed, but I have a big picture question that I think the authors need to consider and at least discuss in the broader context of their study. And this relates to how the authors know that the amount of D-amino acid that they add (and so the amount hydrogen peroxide generated) is relevant to the amounts generated in the context of disease. The intervention leads to systolic dysfunction, but perhaps the hydrogen peroxide concentration, location or duration of production is not relevant to a real world scenario. The authors may have some thoughts on this, and be able to bring some insight to this major question.

Reviewer #2:

Remarks to the Author:

Steinhorn et al. have used a fusion protein, where one domain of the fusion produces hydrogen peroxide and the other senses hydrogen peroxide, to create a new model of cardiac dysfunction. Many conflicting claims about the role of hydrogen peroxide in disease states (causative? something that also happens, but isn't central to the mechanism?) can be found in the literature. This study establishes that elevated hydrogen peroxide alone can cause cardiac dysfunction, and it shows that fibrotic remodeling, while it may occur, is not necessarily an early event.

Novelty:

While using D-amino acid oxidase to elevate peroxide and investigate biological consequences is perhaps new in the heart, similar work by others has been done in other models. It has been reported as protective in some amounts/systems (<http://www.pnas.org/content/107/40/17385> and https://www.sciencedirect.com/science/article/pii/B97801280141580_0014X?via=ihub) and nontoxic (<https://www.liebertpub.com/doi/abs/10.1089/ars.2013.5618>) or toxic (<https://pubs.acs.org/doi/abs/10.1021/acssynbio.6b00120>) in others. Use of this tool to isolate the effects of hydrogen peroxide is not new and past work by others should be cited when the tool is introduced in the third paragraph.

Potential to impact thinking in the field:

The authors have convincingly shown that elevated hydrogen peroxide (+/- D-amino acid oxidase and +/- D-ala or L-ala) can induce cardiac dysfunction. It is less clear whether the elevation here and the elevations observed in disease states are similar. Are the markers of oxidation referenced in the first paragraph (12,13) similar here and in naturally occurring disease states? To support the claims of disease relevance, more comparison with in vivo/ex vivo/other established models is needed.

Technical feedback:

--the western blots should be quantified so that the amount of the fusion protein that was expressed can be reported

Whether D-amino acid induces the effects observed in this study likely depends on expression level of the chemogenetic tool

--pH controls are necessary for HyPer

The controls that were cited in past work were from a very different cell type and system
--some recent work contradicts the statement that D-alanine is absent from mammalian cells (<https://www.frontiersin.org/articles/10.3389/fmolb.2017.00082/full>)
--It is compelling that Fig 4 shows both transcript and protein data. Could Fig. 1 show the same comparison so that the connection between transcript and protein can be more clearly established given that transcript only can sometimes be misleading?
--Is 1 M D-ala in the drinking water correct, or a typo?

Rigor/reproducibility/do others have the necessary details?

--DNA sequence of the fusion construct and control virus should be provided (verified via sequencing)

--More details of imaging and image analysis should be provided (enough for replication) as SI rather than referring the reader to ref. 33.

--Fig. 1

how were the cells chosen for analysis? Given that some cells in the field of view have a fold change of 5 and some seem more similar to a fold change of 3, the size of the error bars seem surprising. Why do some cells respond more than others?

Can a lower magnification image or other fields of view be included in SI?

Why does the 5 mM L-ala curve trend upward significantly in 1B while the curve in 1C for 10 mM L-ala stay flat?

--Fig. 3

Can the authors comment on possible reasons for the animal to animal variability observed in 3C?

--Fig. 4

How were the protein abundances in 4D measured?

Checklist items:

--Data availability statement needed

--Many data plots have all points shown as per the journal's requirements. Some do not.

--Ethics approval/compliance boxes not checked

Dear Dr. Piccoli,

We thank you and the Reviewers for your thoughtful and insightful critiques of our manuscript entitled "Chemogenetic generation of hydrogen peroxide in the heart induces severe cardiac dysfunction." We have performed several additional experiments and have revised the text and provide several new Figures in an effort to comprehensively address the reviewers' criticisms and queries; a point-by-point response to the reviewers' comments follows below. A vertical line in the left margin denotes all places where the text was revised from the original. We feel that our manuscript has been strengthened by the results of these new experiments and by our revisions that respond to the reviewers' thoughtful queries and suggestions.

We are hopeful that with these additional experiments, revisions and clarifications that our manuscript is now acceptable for publication in *Nature Communications*.

Responses to Reviewer 1:

We are pleased by the Reviewer's comment that our study represents a "*stimulating study presenting some very interesting findings*". In the revised manuscript we have tried to respond to the Reviewer's remarks, which are shown *in italics* in their entirety below.

Reviewer #1 (Remarks to the Author):

"This is a stimulating study presenting some very interesting findings, making use of a technology probably only used before in cell systems. The data collected using the model in which peroxide can be measured and inducibly increased by provision of a D-amino acid is and interesting a potentially valuable approach that would be of broad interest to many research fields. As the authors conclude, this work provides evidence that such "chemogenetic" may help in our understanding of redox alteration to disease, in this case heart failure.

"I am confident of the finding and a comprehensive series of experiments have been performed, but I have a big picture question that I think the authors need to consider and at least discuss in the broader context of their study. And this relates to how the authors know that the amount of D-amino acid that they add (and so the amount hydrogen peroxide generated) is relevant to the amounts generated in the context of disease. The intervention leads to systolic dysfunction, but perhaps the hydrogen peroxide concentration, location or duration of production is not relevant to a real world scenario. The authors may have some thought on this, and be able to bring some insight to this major question."

We appreciate the Reviewer's suggestion that we discuss the level of oxidative stress in our model in the context of other heart failure models. While it is difficult to quantify levels of intracellular H₂O₂ *in vitro* let alone *in vivo*, we believe that markers of the overall redox state of the cell can provide some insight into the aggregate level of oxidative stress. To respond to the Reviewer's comment, we performed new experiments to measure the relative redox state of the glutathione pool, and we found that the availability of reduced glutathione (GSH) in the heart was decreased by approximately 50% (new Figure 4C in the revised ms); this value is in line with other models of heart failure, including myocardial infarction and diabetic cardiomyopathy (13,33). Similarly, total levels of glutathione were increased, a change which is also observed with natural cardiac pathologies (21). We have explicitly discussed the Reviewer's observation and the implications of our new data in the Discussion section of the revised manuscript, on page 4.

Responses to Reviewer 2:

We appreciate the Reviewer's observation that in our study we "*have convincingly shown that elevated hydrogen peroxide...can induce cardiac dysfunction*". The reviewer raises several cogent and important criticisms and queries, which are included below and shown *in italics*. We have tried to comprehensively address each of these queries and criticisms in our revised manuscript, as detailed point-by-point below.

Novelty:

While using D-amino acid oxidase to elevate peroxide and investigate biological consequences is perhaps new in the heart, similar work by others has been done in other models. It has been reported as protective in some amounts/systems (<http://www.pnas.org/content/107/40/17385> and <https://www.sciencedirect.com/science/article/pii/B978012801415800014X?via=ihub>) and nontoxic (<https://www.liebertpub.com/doi/abs/10.1089/ars.2013.5618>) or toxic (<https://pubs.acs.org/doi/abs/10.1021/acssynbio.6b00120>) in others. Use of this tool to isolate the effects of hydrogen peroxide is not new and past work by others should be cited when the tool is introduced in the third paragraph.

We regret that we did not sufficiently acknowledge prior work by others on the use of DAAO in other model systems. These previous papers described studies of DAAO in cultured cell systems, whereas our manuscript reports for the first time the successful application of DAAO approaches in a living animal *in vivo*. We feel that the novelty of our model system is in our application of DAAO approaches both in the heart and *in vivo*, and we have updated the manuscript to include a discussion of prior work that exploited DAAO in *in vitro* cell culture systems (references 23, 24 and 26).

Potential to impact thinking in the field:

*The authors have convincingly shown that elevated hydrogen peroxide (+/- D-amino acid oxidase and +/- D-ala or L-ala) can induce cardiac dysfunction. It is less clear whether the elevation here and the elevations observed in disease states are similar. Are the markers of oxidation referenced in the first paragraph (12,13) similar here and in naturally occurring disease states? To support the claims of disease relevance, more comparison with *in vivo/ex vivo*/other established models is needed.*

We appreciate the Reviewer's cogent question about how the level of oxidative stress produced by *in vivo* activation of DAAO compares to other cardiac pathologies. The quantitation of cardiac ROS levels under different physiological and pathological conditions has been difficult to address since there are many factors at play. Furthermore, many of the analytic methods lack specificity and cannot be applied *in vivo* (14, 15). To determine the relative level of oxidant stress in this *in vivo* chemogenetic system, we performed new experiments that measured the relative redox state of the glutathione pool (Figure 4C in the revised manuscript). We found that the level of reduced glutathione (GSH) was decreased by approximately 50% following chemogenetic generation of H₂O₂, which is similar to other disease models of heart failure, including myocardial infarction (33) and diabetic cardiomyopathy (13). We also performed new experiments that revealed that the total levels of glutathione were increased in hearts from animals treated with DAAO (Figure 4D in the revised manuscript), a finding that has also been observed in other models of heart failure (21). Taken together, we feel that our chemogenetic approaches using DAAO have generated levels of redox stress that are similar to levels previously reported in other heart failure models.

--the western blots should be quantified so that the amount of the fusion protein that was expressed can be reported. Whether D-amino acid induces the effects observed in this study likely depends on expression level of the chemogenetic tool

We appreciate the Reviewer's point, and we have performed new experiments to quantitate the relative expression of the HyPer-DAAO fusion protein in individual animals (Supplemental Figure 1). As expected, there was some heterogeneity in the levels of recombinant protein expression between individual animals; however, the levels of DAAO expression were similar in different animals, and this nominal level of between-animal variation in DAAO expression did not allow us to determine whether the relative levels of DAAO expression correlates with the severity of the cardiac phenotype observed in different animals. We now make this point in the revised manuscript, on pages 4-5.

--pH controls are necessary for HyPer. The controls that were cited in past work were from a very different cell type and system

We realize that as a YFP derivative, changes in pH can induce artifactual changes in the fluorescence of HyPer. These artifacts are of particular concern when studying physiologic

signaling where the goal of a fluorescent imaging experiment is to clarify whether small changes in HyPer fluorescence represent changes in H₂O₂ or simply fluctuations in pH. In prior work, we have developed and applied a virus carrying a pH control (SypHer2; reference 41) for this very reason. However, HyPer's main function in our current studies was to serve as a readout of DAAO activity, an enzyme that we have already demonstrated induces no pH change on its own in 2 different mammalian cell types (26). The HyPer results provide supportive evidence that DAAO is functional in cardiac myocytes after infecting animals with the recombinant AAV9 construct expressing the HyPer-DAAO fusion protein, and should be taken in conjunction with the other data presented including induction of oxidative stress transcriptional responses and the new data demonstrating oxidation of the glutathione pool.

--some recent work contradicts the statement that D-alanine is absent from mammalian cells (<https://www.frontiersin.org/articles/10.3389/fmolb.2017.00082/full>)

We appreciate the Reviewer's point, and our revised manuscript now cites some recent reports that D-amino acids are not *entirely* absent from mammalian cells (27). However, we are still confident that, in the heart, D-amino acids are not present at levels sufficient to induce pathology based on our experiments shown in Supplemental Figure 3, in which DAAO-infected animals fed L-alanine showed no effect on cardiac function (Supplementary figure 4). We have revised the text both to clarify this point and to acknowledge the presence of D-amino acids in some mammalian tissues.

--It is compelling that Fig 4 shows both transcript and protein data. Could Fig. 1 show the same comparison so that the connection between transcript and protein can be more clearly established given that transcript only can sometimes be misleading?

We agree that changes in transcript level can be misleading since concomitant changes in post-transcriptional regulatory mechanisms may have a significant independent effect on total protein levels. In our revised manuscript, we now discuss this important consideration (page 4). We also point out that our goal in Figure 1 was to probe the *activation* of redox-sensitive transcription factors rather than to assess the overall downstream consequences (i.e. effectiveness of protein translation). The oxidation events responsible for the activation of Nrf2 and NF-κB are not themselves easily measured, so we chose to assess the most proximal measure of changes in their activity, specifically, transcript abundance. Furthermore, the experiments in Figure 1 take place over 2 hours, a time span that is likely too short to reliably detect changes in protein abundance.

--Is 1 M D-ala in the drinking water correct, or a typo?

The rats were indeed fed 1 M D-alanine, which had no independent effects on the health of the animals including body weight or basic metabolic parameters. In our revised manuscript, we have included these data in a new supplemental figure 7. In order to control for other potential independent effects of D-alanine, the control animals were also fed D-alanine at the same 1 M concentration. Since there are no stereospecific mammalian transporters for D-alanine, we anticipated that the D-alanine provided in the drinking water would be poorly absorbed, and we wanted to be sure that we would be able to provide sufficient substrate for the DAAO expressed in cardiac myocytes.

Rigor/reproducibility/do others have the necessary details?

--DNA sequence of the fusion construct and control virus should be provided (verified via sequencing)

In the revised manuscript, we have included the DNA sequence for both the fusion construct and control virus, both of which were verified by sequencing, now shown in the supplementary material.

--Fig. 1: how were the cells chosen for analysis? Given that some cells in the field of view have a fold change of 5 and some seem more similar to a fold change of 3, the size of the error bars

seem surprising. Why do some cells respond more than others?

In our revised manuscript, we have added additional details on the imaging and image analysis. In the revised manuscript, we have further expanded the Methods section to explain how cells were chosen for analysis. We attribute the relatively small standard error bars to the relatively high n for each experiment ($n \sim 30$ in figure 1B). We believe that the variability in responses is attributable to heterogeneity of HyPer-DAAO expression from myocyte to myocyte. In myocytes expressing less recombinant protein, less H_2O_2 is produced for the same D-alanine stimulus, thereby causing a smaller increase in the HyPer ratio. The greater increase in HyPer ratio in myocytes with more HyPer-DAAO expression (myocytes which appear brighter independent of any color change) is consistent with this explanation.

Can a lower magnification image or other fields of view be included in SI?

Due to the relatively low numerical aperture of the 10X objective available to us, we have only quantitated fluorescence images at 20X. In response to the Reviewer's query, we have included additional fields of view in a new Supplemental Figure 2.

Why does the 5 mM L-ala curve trend upward significantly in 1B while the curve in 1C for 10 mM L-ala stay flat?

The most likely explanation for the qualitative difference in the control curves is that the two experiments occurred over different timescales, with more time available for the curve in Figure 1B to drift upwards. A retrospective examination of the data revealed that there did appear to be a population of myocytes from a single control experiment that contributed to Figure 1B that drifted upwards, resulting in the average curve also drifting upwards. However, we could find no valid reason to exclude this control experiment from our pooled results. We are still confident in our conclusions from these fluorescence traces since all statistical analyses are a comparison between data sets that were acquired with contemporaneous controls, and the difference is highly significant.

--Fig. 3 Can the authors comment on possible reasons for the animal to animal variability observed in 3C?

We attribute the animal-to-animal variations in 3C in part to variable expression of the HyPer-DAAO fusion protein, and we have performed immunoblots on samples from these same animals to look at differences in HyPer-DAAO expression (see point 3 above). A second likely reason for the animal-to-animal variability may represent changes occurring after the hearts were ischemic, but before they were snap frozen for immunoblot analyses. Phospholamban phosphorylation is tightly and rapidly regulated, and variation in the time to dissect, explant and mince the hearts (during which time they are ischemic) might produce variable alterations in phospholamban phosphorylation state. We now discuss this point in the revised manuscript (page 4).

--Fig. 4 How were the protein abundances in 4D measured?

We regret that we were not clearer in our description of the data presented in 4D (this is now panel 4B in the revised manuscript), and we have revised the figure legend to clarify this point. These data represent transcript abundance measured by qPCR and are intended to mirror the *in vitro* experiments from Figure 1 examining activation of redox-sensitive transcriptional programs.

Checklist items:

--Data availability statement needed

--Many data plots have all points shown as per the journal's requirements. Some do not.

--Ethics approval/compliance boxes not checked

All checklist items are now completed and checked off: data availability statement, all data points are now shown, animal protocol ethical approval statement included in the body of the ms.

Dear Dr. Piccoli,

We thank you and the reviewers again for your thoughtful and insightful critiques of our manuscript entitled "Chemogenetic generation of hydrogen peroxide in the heart induces severe cardiac dysfunction." We are pleased that reviewer 1 recommends publication and appreciate reviewer 2's concern regarding possible effects of pH on our fluorescence time-course experiments. We have therefore performed additional control experiments using cardiac myocytes derived from human IPS cells to clearly show that the changes in fluorescence we observe in cardiac myocytes are due to hydrogen peroxide rather alterations in pH. We have also attempted to clarify how the look-up tables for the ratiometric images in Figure 1 were generated. We are hopeful that with these additional experiments, revisions and clarifications that our manuscript is now acceptable for publication in *Nature Communications*.

Reviewer 2:

- 1) "The authors addressed the reviews substantively. One issue was not addressed in a satisfactory manner, however. Establishing the use of HyPer-DAAO in the heart of living animals to study pathophysiology induced by increased levels of hydrogen peroxide is the central aim of the study. To establish the chemogenetic tool in this context in a rigorous way, the pH control should be done.

In reference 26, the cell types were HeLa and NIH-3T3, which are quite different than cardiac cells. To establish the tool in this new cell/tissue/organ/whole animal, controls should be rigorous.

There are some inter-related issues that make this control central to making the claims of the paper unassailable:

--HyPer ratios increase rather dramatically with increasing pH, and the difference between ratio in the reduced state and the oxidized state lessens. Knowing if the ratio change is all due to H₂O₂ or has some pH component is valuable information.

--The tool makes byproducts that could potentially alter the pH
As the authors note, it is difficult to quantitate H₂O₂ in vivo. The arguments about basal NH₃ are completely reasonable, but due to the difficulty in quantitating H₂O₂ we do not really know how high those levels are going and how much NH₃ and imino acid byproducts are being produced. It could be quite a lot to change the total GSH and GSH/GSSG balance since GSH is mM.

Without repeating the whole animal study, is it possible to do any kind of SypHer-DAAO control in a cardiac cell?"

We appreciate the reviewer's point that DAAO produces products that may increase intracellular pH and spuriously increase the HyPer ratio. We therefore sought to identify a tractable cardiac cell in which to validate the HyPer signal. We studied cardiac myocytes derived from human induced-pluripotent stem cells (iPSCs), chosen as an informative and tractable cardiac cell in which to perform these key pH control experiments. We performed fluorescence time-course experiments in iPSC-derived cardiac myocytes that we transfected with plasmids expressing either SypHer2-DAAO or HyPer-DAAO. SypHer2 is identical to HyPer except that one of the cysteine residues is changed to serine, preventing disulfide formation by hydrogen peroxide; thus, any

change in fluorescence with SypHer2 would *not* be due to hydrogen peroxide, and would reflect a pH effect. These data are now presented as a new Supplemental Figure 3. We once again observed a robust rise in the HyPer ratio with the addition of D-alanine, while the fluorescence ratio in cells expressing SypHer2 did not increase with D-alanine. We feel that these new control experiments support our claim that the physiologic effects of DAAO activation are due to H₂O₂ rather than a possible pH change, and believe these results– which control for HyPer’s pH sensitivity– strengthen our interpretation of the HyPer-DAAO imaging experiments shown in Figure 1.

2) “Much more minor:

It could be confusing that the HyPer ratios go from 0-1.2 in Fig 1a and from 1-4 in Fig 1b and 1c. How to get a ratio of 0 (1a) and the normalization procedure (1b and 1c) are unclear from the description in the caption and the methods section.”

We regret that we did not clearly describe how color lookup tables for ratiometric images and fluorescence time traces are displayed. The color lookup table for Figure 1A maps the ratio of each pixel to the color in the colorbar, and maps the intensity of the denominator image to a luminance value. The pseudocolor images in Figure 1a display the raw HyPer ratio after background subtraction. However, all ratiometric fluorescence traces displayed in the manuscript depict ratios that have been normalized to the ratio prior to the start of treatment. Therefore, the increase in normalized ratio from 1 to 4 in Figure 1B reflects an increase in the raw HyPer ratio from ~0.25 to 1.0 in the images shown in Figure 1A. We realize that the somewhat compressed color range at the lower end of the lookup table may have made it difficult to discriminate between ratios below 0.25 and given the false impression that the raw HyPer ratio of those cells was 0. We have clarified these points in the Methods section and in the figure legend. We have also further modified the color lookup table to aid readers in discriminating between different ratios.